# Neonatal antibiotic exposure impairs child growth during the first six years of life by perturbing intestinal microbial colonization

Atara Uzan-Yulzari[1,16], Olli Turta[2,16], Anna Belogolovski[3], Oren Ziv[1], Christina Kunz[4], Sarah Perschbacher[5], Hadar Neuman[1,14], Edoardo Pasolli[6], Aia Oz[7], Hila Ben-Amram[1,15], Himanshu Kumar[8], Helena Ollila [9,10], Anne Kaljonen[9], Erika Isolauri [2], Seppo Salminen[8], Hanna Lagström[10], Nicola Segata [6], Itai Sharon[7,11], Yoram Louzoun [3], Regina Ensenauer[5,12], Samuli Rautava [2,13,16 ✉] & Omry Koren [1,16 ✉]

Exposure to antibiotics in the first days of life is thought to affect various physiological aspects of neonatal development. Here, we investigate the long-term impact of antibiotic treatment in the neonatal period and early childhood on child growth in an unselected birth cohort of 12,422 children born at full term. We find significant attenuation of weight and height gain during the first 6 years of life after neonatal antibiotic exposure in boys, but not in girls, after adjusting for potential confounders. In contrast, antibiotic use after the neonatal period but during the first 6 years of life is associated with significantly higher body mass index throughout the study period in both boys and girls. Neonatal antibiotic exposure is associated with significant differences in the gut microbiome, particularly in decreased abundance and diversity of fecal Bifidobacteria until 2 years of age. Finally, we demonstrate that fecal microbiota transplant from antibiotic-exposed children to germ-free male, but not female, mice results in significant growth impairment. Thus, we conclude that neonatal antibiotic exposure is associated with a long-term gut microbiome perturbation and may result in reduced growth in boys during the first six years of life while antibiotic use later in childhood is associated with increased body mass index.

[1] Azrieli Faculty of Medicine, Bar Ilan University, Safed, Israel. [2] Department of Pediatrics, University of Turku and Turku University Hospital, Turku, Finland. [3] Department of Mathematics, Bar-Ilan University, Ramat-Gan, Israel. [4] Institute of Child Nutrition, Max Rubner-Institut, Federal Research Institute of Nutrition and Food, Karlsruhe, Germany. [5] Institute for Medical Information Processing, Biometry, and Epidemiology (IBE), Faculty of Medicine, Ludwig-Maximilians-Universität München, Munich, Germany. [6] Department CIBIO, University of Trento, Trento, Italy. [7] Migal – Galilee Research Institute, Qiryat Shemona, Israel. [8] Functional Foods Forum, University of Turku, Turku, Finland. [9] Department of Biostatistics, Faculty of Medicine, University of Turku, Turku, Finland. [10] Department of Public Health, University of Turku and Turku University Hospital, Turku, Finland. [11] Tel-Hai Academic College, Qiryat Shemona, Israel. [12] Department of General Pediatrics, Neonatology and Pediatric Cardiology, University Children's Hospital, Heinrich Heine University Düsseldorf, Düsseldorf, Germany. [13] University of Helsinki & Helsinki University Hospital, New Children's Hospital, Pediatric Research Center, Helsinki, Finland. [14]Present address: Zefat Academic College, Safed, Israel. [15]Present address: Ziv Medical center, Safed, Israel. [16]These authors contributed equally: Atara Uzan-Yulzari, Olli Turta, Samuli Rautava, Omry Koren. ✉email: samuli.rautava@hus.fi; omry.koren@biu.ac.il

Newborn infants are highly susceptible to invasive bacterial infections, and suspected infection is one of the most common reasons for admission to a neonatal unit. The incidence of blood culture-confirmed early-onset neonatal sepsis ranges between 0.5 and 0.8 in 1000 newborns[1,2] and culture-negative clinical sepsis has been estimated to be 6–16 times more frequent[2]. According to current recommendations[1], all neonates with clinical signs suggesting bacterial infection, in addition to selected high-risk infants, receive empirical antimicrobial therapy. Consequently, a considerable number of neonates are treated with antibiotics during the first days of life[3,4].

Neonates subjected to antibiotic therapy reportedly exhibit altered gut microbiome composition during the first weeks of life[5], but the clinical or microbiological long-term consequences of this exposure remain unknown. Given the causal links between the intestinal microbiome and growth, obesity, and metabolic disease[6,7], together with data from epidemiological[8,9] and experimental[10,11] studies, it has been suggested that neonatal antibiotic exposure might have an impact on growth in early life[12]. Recently, antibiotic therapy in the first week of life was reported to be associated with decreased growth during the first year of life[13], but longer-term outcomes remain unknown. Later in infancy and childhood, antibiotic use has been linked to increased risk of becoming overweight and obese[8,9,12]. We hypothesized that antibiotic treatment during the first days of life may exert a long-lasting effect on childhood growth by disrupting the natural gut microbial colonization process.

In this work, we investigated the association between antibiotic exposure during and after the neonatal period and child growth until the age of six years in a large unselected cohort and validated our findings in an independent cohort. We show that neonatal antibiotic exposure is associated with reduced weight and height gain during the first six years of life in boys but not in girls. After the neonatal period, antibiotic use is associated with increased body mass index (BMI) in both boys and girls. Experimental studies using germ-free mice demonstrate that fecal microbiota transplant (FMT) with fecal samples from infants exposed to antibiotics in the neonatal period results in growth failure in male but not in female mice, which suggests a causal role for the antibiotic-induced gut microbiome perturbations in the pathogenesis of the growth impairment in antibiotic-exposed boys.

## Results

**Neonatal antibiotic exposure and child growth.** The impact of neonatal antibiotic exposure on growth during the first six years of life was first investigated in the Southwest Finland Birth Cohort (SFBC) consisting of all 14,969 children born at the Turku University Hospital in Turku, Finland during the years 2008–2010. Altogether 12,422 children born from singleton pregnancies at full term (after $36^{6/7}$ weeks of gestation) who did not have genetic abnormalities or significant chronic disorders affecting growth and who did not need long-term prophylactic antibiotic treatment were included in this study. Antibiotics had been administered within the first 14 days of life to 1,151 (9.3%) of the neonates in the study. A vast majority of the subjects were treated with a combination of intravenous benzylpenicillin and gentamicin as per the hospital guidelines. To differentiate between the impact of antibiotic exposure and the underlying infection, antibiotic treatment was categorized to brief empirical antibiotic treatment, which was discontinued after infection had been ruled out (513 neonates or 4.1% of the total study population) and to antibiotics administered for confirmed or clinical infection (638 neonates or 5.1% of the study population). The background and clinical characteristics of the children are presented in detail in Table 1. Previous epidemiological[8,14] and

experimental[10] studies suggest sexual dimorphism in the susceptibility to growth disturbances after early-life antibiotic exposure. Furthermore, boys were significantly more often exposed to neonatal antibiotics (Supplementary Table 1) and, consequently, boys and girls were analyzed separately. To control for potential confounding factors which might be associated with neonatal antibiotic exposure and affect childhood growth, gestational age, birth weight Z-score, mode of delivery, maternal prepregnancy BMI, and intrapartum antibiotic treatment were included in the hierarchical linear mixed model for repeated measurements as explanatory variables with neonatal antibiotic exposure.

In the adjusted model, boys exposed to brief empirical antibiotic exposure or antibiotic treatment for infection exhibited significantly lower weight (Fig. 1a) compared to non-exposed children throughout the first six years of life ($p = 0.007$ for infected boys, and $p = 0.031$ for non-infected boys). Boys exposed to antibiotic treatment for infection also exhibited significantly lower height ($p = 0.011$, Fig. 1b) and BMI between the ages of 2 and 6 years ($p = 0.037$, Supplementary Fig. 1a). No statistically significant association between neonatal antibiotic exposure and weight (Fig. 1c), height during the first six years of life (Fig. 1d), nor BMI (Supplementary Fig. 1b) were observed in girls after adjusting for the same potential confounders.

We next proceeded to confirm this finding in the independent German mother-child cohort Programming of Enhanced Adiposity Risk in Childhood–Early Screening (PEACHES). The PEACHES cohort consists of 1,707 offspring and their mothers recruited during pregnancy between 2010 and 2015[15,16]. For analysis of the effect of neonatal infection treated with antibiotics on child weight and height from birth to 5 years, we included mother-child dyads with maternal prepregnancy BMI < 30 kg/m², singleton pregnancies at full term and offspring without chronic disease. Within the analyzed population ($n = 535$), 6.4% (girls: 4.3%, boys: 8.5%) of the neonates had an infection and antibiotic therapy. Relevant clinical characteristics, as mean with standard deviation (SD) or %, included birth weight Z-score (girls: 0.2, SD 0.9, boys: 0.3, SD 0.9), maternal pre-pregnancy BMI in kg/m² (girls: 24.0, SD 3.3, boys: 23.9, SD 3.1), vaginal delivery (girls: 68.2%, boys: 58.9%) and gestational age in weeks (girls: 39.6, SD 1.1, boys: 39.5, SD 1.1). No differences in breastfeeding rates were detected between infants with neonatal infection and antibiotic treatment and non-infected infants during the first 6 months of life (Supplementary Table 2). Boys but not girls exposed to antibiotic therapy in the neonatal period exhibited significantly lower weight and height Z-scores during the first five years of life as compared to non-exposed children (Table 2). A hierarchical linear mixed model for repeated measurements adjusted for gestational age, birth weight Z-score, mode of delivery, child's age, maternal prepregnancy BMI, intrapartum antibiotic treatment, any breastfeeding during the first 6 months of life and the interaction between neonatal infection and child's age was applied. No differences in BMI Z-scores were detected.

**Antibiotic use after the neonatal period and child growth.** In contrast to our results indicating reduced childhood weight and height gain in boys treated with antibiotics in the first days of life, previous reports suggest that antibiotic exposure later in infancy may be associated with increased risk of overweight and obesity[8,9,14]. We, therefore, wanted to investigate whether the association between antibiotic use and child growth was dependent on the age at which the exposure occurs. Data regarding antibiotic purchases for the children in the SFBC were obtained from the Drug Prescription Register maintained by the Social Insurance Institution of Finland. The cumulative number of antibiotic

**Table 1 Clinical characteristics of the children in the Southwest Finland Birth Cohort (SFBC).**

| | Boys | | | | Girls | | | |
|---|---|---|---|---|---|---|---|---|
| | No neonatal antibiotic exposure | Brief empirical antibiotic exposure | Antibiotic treatment for infection | p | No neonatal antibiotic exposure | Brief empirical antibiotic exposure | Antibiotic treatment for infection | p |
| Subjects | 89.3% (5643/6316) | 4.7% (297/6316) | 6.0% (376/6136) | | 92.2% (5637/6115) | 3.5% (216/6115) | 4.3% (262/6115) | |
| Gestational age (weeks) | 40.0 (1.2) | 40.11 (1.38) | 40.06 (1.34) | 0.49 | 40.10 (1.18) | 39.98 (1.45) | 40.10 (1.30) | 0.34 |
| Birth weight (grams) | 3625 (462) | 3680 (530) | 3754 (499) | <0.0001 | 3509 (439) | 3509 (602) | 3614 (512) | 0.0011 |
| Birth weight Z score | 0.01 (1.03) | 0.13 (1.18) | 0.29 (1.11) | <0.0001 | −0.01 (1.04) | −0.02 (1.43) | 0.24 (1.20) | 0.0014 |
| Maternal prepregnancy BMI[a] | 24.3 (4.7) | 25.2 (5.2) | 25.2 (5.4) | 0.003 | 24.3 (4.7) | 25.0 (5.0) | 24.9 (4.7) | 0.032 |
| Vaginal delivery | 88% (4941/5634) | 83% (246/297) | 83% (313/376) | <0.0001 | 88% (4959/5637) | 77% (166/216) | 87% (229/262) | <0.0001 |
| Intrapartum antibiotic exposure | 10% (571/5619) | 16% (48/296) | 17% (63/375) | 0.32 | 10% (588/5614) | 9% (20/216) | 18% (48/262) | 0.0003 |
| Antibiotic prescriptions until six years of age[b] | 7 (4, 12) | 7 (3, 11) | 7 (4, 13) | | 6 (3, 10) | 6 (3, 11) | 7 (3, 11) | 0.23 |
| Age at neonatal antibiotic therapy (hours)[c] | — | 5.5 (1.5, 24.9) | 4.1 (1.8, 18.5) | 0.56 | — | 3.5 (1.4, 14.1) | 3.7 (1.5, 16.4) | 0.81 |
| Neonatal antibiotic exposure duration (days)[c] | — | 2 (2, 2) | 7 (7, 7) | <0.0001 | — | 2 (2, 2) | 7 (7, 7) | <0.0001 |

The data are presented as means (standard deviation) for continuous variables and as percentage (proportion) for categorical variables. Two-tailed one-way Anova was used for continuous variables and Chi-squared test for categorical variables.
[a]Variable was tested with two-sided Kruskal–Wallis test because of non-normal distribution.
[b]Median (quartiles Q1, Q3) and two-sided Kruskal–Wallis test were used because of non-normal distribution.
[c]Median (quartiles Q1, Q3) and two-sided Wilcoxon rank-sum test were used because of non-normal distribution.

courses after the neonatal period but during the first six years of life was associated with significantly higher BMI Z-scores throughout the study period in both boys (Fig. 2a, p < 0.001) and girls (Fig. 2b, p < 0.001) in a hierarchical linear mixed model for repeated measurements adjusted for gestational age, birth weight Z-score, mode of delivery, maternal prepregnancy BMI and neonatal antibiotic exposure. The number of antibiotic purchases was also associated with significantly higher weight Z-scores during the first six years of life in boys (Supplementary Fig. 2a, p < 0.001) but not in girls (Supplementary Fig. 2c, p = 0.71). Height development during the first six years of life was not associated with antibiotic use after the neonatal period in either boys (Supplementary Fig. 2b) or girls (Supplementary Fig. 2d).

**Neonatal antibiotic exposure and gut microbiome development.** Perturbations of the gut microbiome resulting from antibiotic exposure provide a potential causal mechanism between neonatal antibiotic exposure and reduced childhood growth. The long-term impact of neonatal antibiotic treatment on the gut microbiome was investigated in subjects selected from a clinical trial conducted at Turku University Hospital[17]. Altogether 13 neonates had been exposed to antibiotic therapy with intravenous benzylpenicillin and gentamicin during the first 48 h of life. Twenty full-term healthy neonates not exposed to antibiotics in the neonatal period from the same study were chosen as controls. The subjects exposed and not exposed to neonatal antibiotics were similar with regard to duration of gestation, birthweight, mode of birth, exposure to antibiotics during delivery or the first six months of life after the neonatal period, and breastfeeding at 1 month of age (Supplementary Table 3). However, the mean duration of breastfeeding was shorter in the infants exposed to neonatal antibiotics. Fecal samples were collected at the ages of 1, 6, 12, and 24 months, and the fecal microbiome was analyzed by sequencing of the 16 S rRNA gene and metagenomics for a subset of the children.

Neonatal antibiotic exposure was associated with significant alterations in the gut microbiota throughout the study period. Weighted and Unweighted UniFrac analysis showed a gradual change between 1–24 months in bacterial composition, reflecting the maturation of the gut microbiome in the children not exposed to neonatal antibiotics (Fig. 3a, c, respectively). The subjects exposed to neonatal antibiotics also demonstrated an expansion of bacterial establishment over time but in a distinct and less organized pattern (Fig. 3b, d). To test for statistically significant effects of neonatal antibiotics on the microbiome that occurred consistently, and persisted over 24 months, we projected the data into a lower dimension following log transformation and Z scoring using a 5-dimensional Principal Coordinate Analysis (PCoA). We next performed a Hotelling test on the first 3 dimensions of the PCoA projection among groups (Supplementary Table 4). Significant differences between the gut microbiome of antibiotic-treated and control groups were observed after 1 (p < 0.002), and 6 months (p < 0.003), demonstrating the persistence of the effect of antibiotic exposure on the microbiome (Supplementary Table 4).

The infants exposed to neonatal antibiotics exhibited significantly lower gut microbiome richness as compared to the non-exposed infants at the age of 1 month. Interestingly, at the age of 6 months, the antibiotic-treated infants reached the bacterial richness level of the control infants, and at the ages of 12 and 24 months, the antibiotic-treated subjects gained significantly higher levels of bacterial richness as compared to the control subjects (Fig. 3e).

Taxonomic analysis, based on the relative abundance at the genus level, revealed that the genus *Bifidobacterium* was most

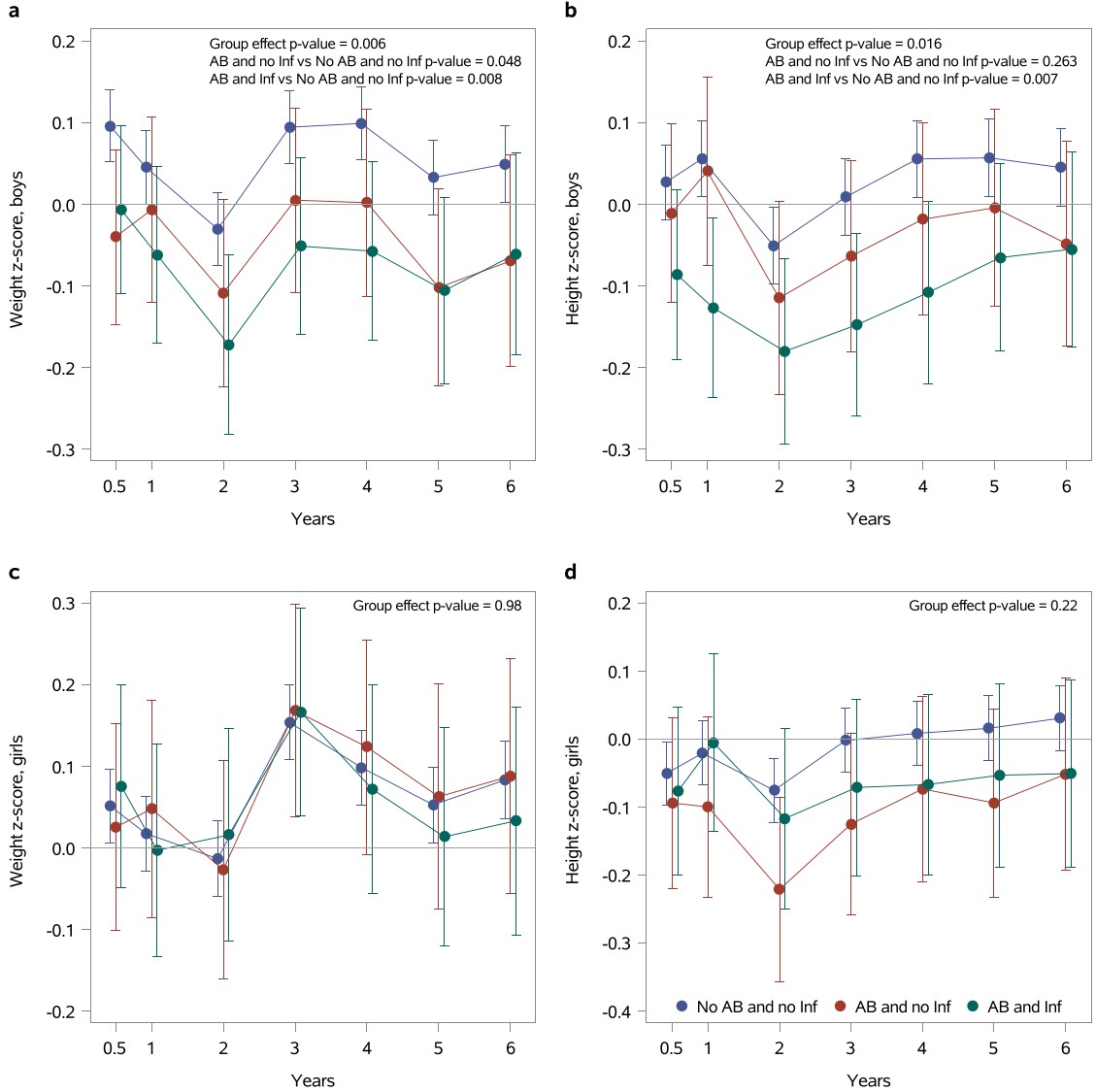

**Fig. 1 Neonatal antibiotic exposure is associated with impaired weight and height gain in boys.** Growth during the first six years in life in children exposed to brief empirical antibiotic therapy (AB and no Inf, $N = 513$) and children who had received antibiotics for confirmed or clinical bacterial infection (AB and Inf, $N = 638$) as compared to children not exposed to antibiotics (No AB and no Inf, $N = 11,271$). Estimates for weight and height Z-scores are presented separately for boys (Fig. 1a Fig. 1b, respectively) and for girls (Fig. 1c Fig. 1d, respectively). The x-axis represents the age in years; the y axis represents the model-based Least Squares Mean (LSM) estimates. The whiskers represent 95% confidence intervals. The data were analyzed using a hierarchical linear mixed model for repeated measurements. Neonatal antibiotic exposure, gestational age, birth weight Z-score, mode of delivery, child's age, maternal prepregnancy BMI, and intrapartum antibiotic treatment were included in the model as explanatory variables.

**Table 2 Weight, height and body mass index (BMI) Z-scores during the first five years of life in children of the Programming of Enhanced Adiposity Risk in Childhood–Early Screening (PEACHES) cohort with neonatal infection and antibiotic therapy vs. children without neonatal infections and in the absence of antibiotic therapy.**

|  | Boys (n = 258) | | Girls (n = 277) | |
| --- | --- | --- | --- | --- |
| Outcome | β-estimate (95% CI) | p | β-estimate (95% CI) | p |
| Weight Z-score | −0.43 (−0.86, −0.002) | 0.0488 | 0.28 (−0.21, 0.77) | 0.2576 |
| Height Z-score | −0.62 (−1.08, −0.16) | 0.0080 | −0.26 (−0.77, 0.26) | 0.3331 |
| BMI Z-score | −0.11 (−0.68, 0.47) | 0.7143 | 0.62 (−0.19, 1.44) | 0.1315 |

β-estimates for weight, height, and BMI Z-scores with 95% confidence interval (CI) and p-values for t-statistics are presented for children of age 0.5–5 years after neonatal infection and antibiotic therapy vs no neonatal infection, grouped by offspring sex. Data is based on hierarchical linear mixed model for repeated measurements adjusted for gestational age, birth weight Z-score, mode of delivery, child's age, maternal prepregnancy BMI, intrapartum antibiotic treatment, and any breastfeeding during the first 6 months of life. Additionally, an interaction term between neonatal infection and child's age is included. Only children with complete covariate sets and at least one BMI Z-score between 6 months and 5 years are presented.

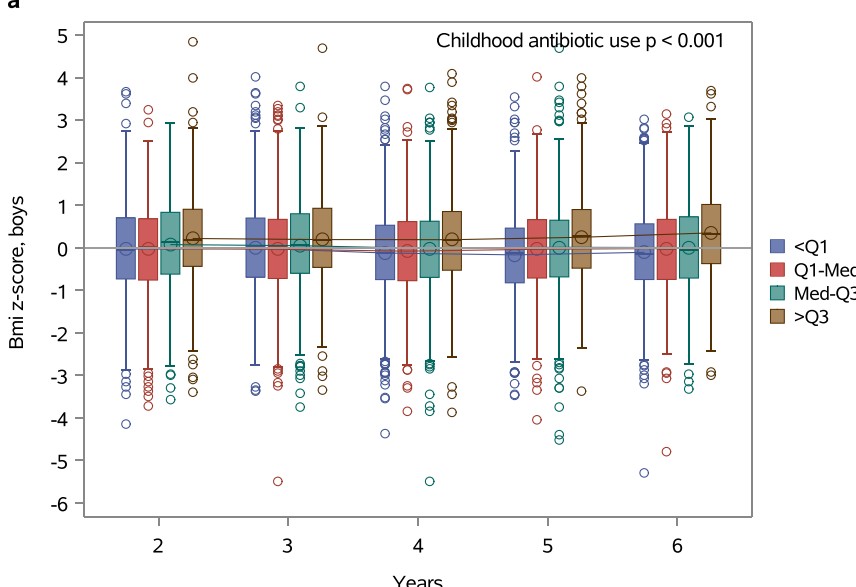

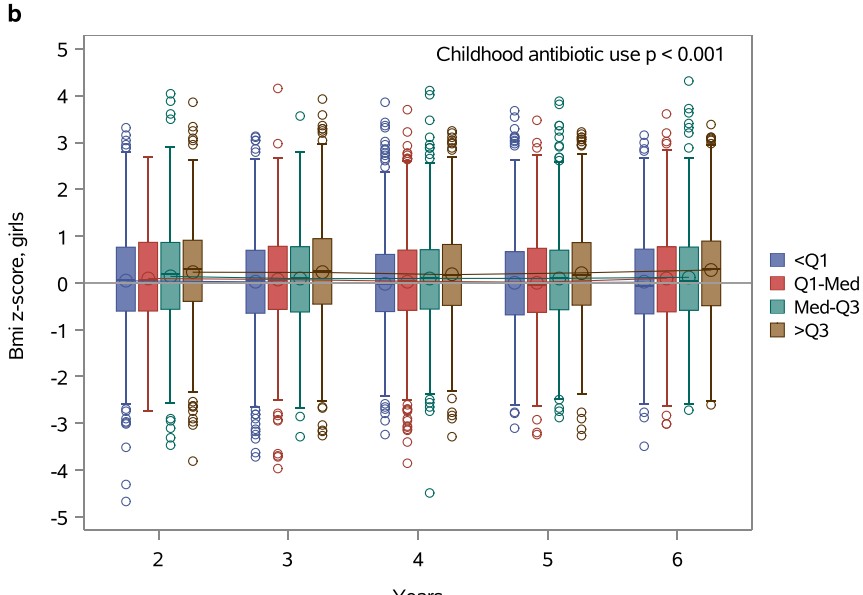

**Fig. 2 Early childhood antibiotic exposure is associated with increased BMI.** The association between childhood antibiotic use and BMI Z-scores during the first six years of life in boys (**a**) and girls (**b**). The subjects have been categorized by quartiles (Q1, median, and Q3) based on the cumulative number of antibiotic purchases at each point in time. The number of antibiotic purchases was associated with significantly higher BMI Z-scores during the first six years of life in both boys ($p < 0.001$) and girls ($p < 0.001$) in a hierarchical linear mixed model for repeated measurements adjusted for gestational age, birth weight Z-score, mode of delivery, maternal prepregnancy BMI and neonatal antibiotic exposure. The boxes represent interquartile range (IQR) and the whiskers represent 1.5 times IQR. The circles represent outliers.

substantially affected by antibiotic treatment. Strikingly, the effect persisted to the age of 24 months (Fig. 3f–h). *Bifidobacterium* is highly abundant throughout infancy, especially during lactation, and significantly decreases thereafter as infants are weaned and the diet expands to solid food[18]. As expected, we found Bifidobacteria to comprise ~50% of total sequences at both 1 and 6 months of age, while decreasing to ~25–30% at 12 and 24 months in the infants not exposed to neonatal antibiotics (Fig. 3f–h). The relative abundance of Bifidobacteria was lower in the infants exposed to neonatal antibiotics as compared to the control infants at all time points, except for 6 months after antibiotic treatment (Fig. 3f–h). Examination at the feature level revealed that the *Bifidobacterium* genus included five unclassified

features of *Bifidobacterium*. The antibiotic-treated infants were shown to be less diverse in fecal *Bifidobacterium* features than the control at all four time points. For instance, the elevation in *Bifidobacterium* in the antibiotic-treated infants after 6 months occurred mainly in a single feature (unclassified1). The features termed unclassified2 and unclassified3 were present mostly in the control infants (Fig. 3g). The differences between the other bacterial genera can be observed in Fig. 3h which summarizes the most significant changes in the gut microbiome between the antibiotic-treated and control subjects at each timepoint. These results indicate that neonatal antibiotic exposure has a long-lasting detrimental impact on intestinal Bifidobacteria, which is detectable even 24 months after exposure.

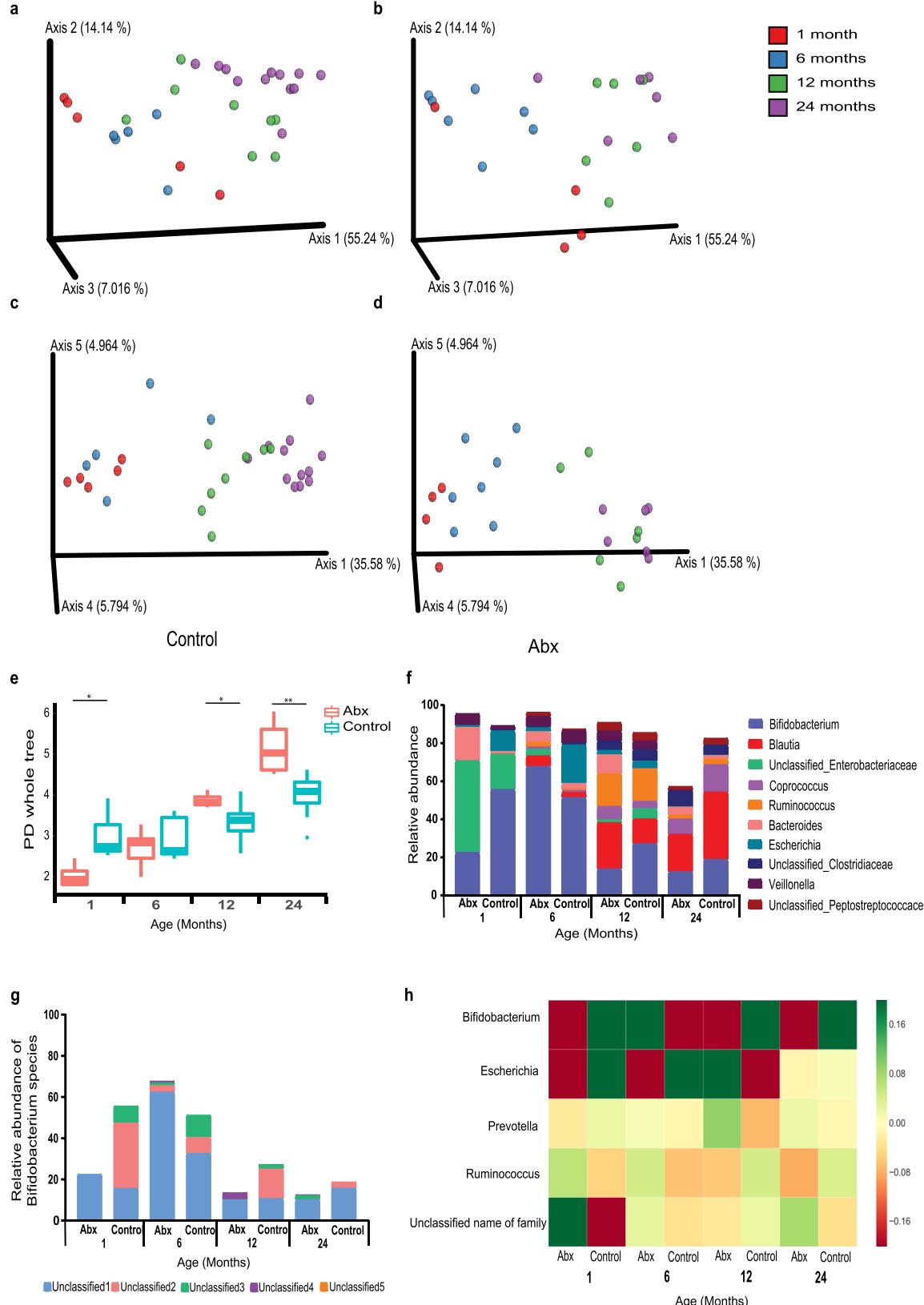

Genome-resolved metagenomics analysis of 27 samples (Supplementary Tables 5 and 6) from 20 infants at the ages of 1 and 6 months revealed six *Bifidobacterium* species that were present in abundance of 0.1% of the community or higher: *B. longum* (appeared in 18 samples), *B. bifidum* (11), *B. breve* (10), *B. dentium* (6), *B. pseudocatenulatum* (4), and *B. adolescentis* (3)

(Supplementary Table 7). At the age of one month, when all of the infants were breastfed, *Bifidobacterium* species were detected almost exclusively in the control group (13 species/$n = 7$, compared to one species in the antibiotics group, $n = 5$, $p$-value$=5.3 \times 10^{-4}$). The differences in presence were most notable for *B. bifidum* (4/7 in the control group, 0/5 in the

**Fig. 3 Alterations in gut bacterial colonization in infants following antibiotic exposure.** 16 S rRNA sequencing was performed to characterize bacterial changes. **a**, **b** Principal Coordinates Analysis (PCoA) based on Weighted UniFrac and (**c**, **d**) Unweighted UniFrac distance matrices in (**a**, **c**) control ($n = 20$), and (**b**, **d**) antibiotic-treated ($n = 13$) infants after 1 (red), 6 (blue), 12 (green), and 24 (purple) months from antibiotic exposure. (**e**) Alpha diversity comparison based on phylogenetic diversity. Significant differences in bacterial richness between control and antibiotic-treated groups are seen at 1 ($p = 0.014$), 12 ($p = 0.014$) and 24 ($p = 0.001$) months following antibiotic exposure. (**f**) Relative taxonomic composition based on 10 most abundant genera of control and antibiotic-treated groups. (**g**) Relative levels of *Bifidobacterium* species. (**h**) Average log10 change in the bacteria showing significant differences between groups ($p < 0.05$ and FDR correction). Green values imply over-representation, and red represents under representation. (*$p < 0.05$, **$p < 0.01$, ***$p < 0.001$ and ****$p < 0.0001$; minimum four samples in each group).

antibiotics group) and *B. longum* (4/7 in the control group, 1/5 in the antibiotics group). These species, along with *E. coli* and *V. parvula*, exhibited the most notable differences between the control and antibiotics groups of all 88 bacterial species detected with abundance >0.1% in the one-month samples (Supplementary Table 8). No significant differences in the presence of *Bifidobacterium* species were observed in the six-month samples (22 species in the control group, $n = 9$, compared to 16 species in the antibiotics group, $n = 6$, $p$-value=0.46); however, difference in *B. bifidum* presence was notable (6/9 in control samples, 1/6 in antibiotics samples, see Supplementary Fig. 3).

**Fecal microbiota transplant to germ-free mice.** To establish whether causal relationships exist between neonatal antibiotic exposure, long-term gut microbiota perturbations, and reduced growth, we next used fecal microbiota transplantation (FMT) from antibiotic-exposed and non-exposed infants (Supplementary Table 9) to germ-free (GF) mice. A significant reduction in relative weight gain was observed in male mice that received FMT from infants 1 month after antibiotic treatment during the 43-day follow-up period, as compared to male mice that received FMT from non-exposed infants (Fig. 4a). In contrast, FMT with feces obtained from antibiotic-exposed children at 1 month of age did not affect the growth of female mice (Fig. 4b). It is of note that growth impairment was observed in male mice also when FMT was performed with samples collected 24 months after antibiotic exposure (Supplementary Fig. 4a). These data suggest that the reduced growth in boys and the long-term perturbations in the gut microbiome after neonatal antibiotic exposure are causally linked.

We next analyzed the bacterial composition of the mice receiving FMT from antibiotic-treated and control infants 1 month and 24 months from antibiotic exposure. FMT from 1-month old infants not exposed to antibiotics resulted in continuous elevation in bacterial richness from day 3 after FMT until the end of the experiment. In contrast, mice receiving feces from infants 1 month after neonatal antibiotic exposure demonstrated an increase in bacterial richness only from day 3 to day 7, and thereafter the richness remained constant. Consequently, significant differences in bacterial richness between the two groups were detected at four-time points: day 14, day 21, day 35, and day 43 (Fig. 4c). No differences were seen in the microbiota of mice receiving FMT using the 24-month samples (Supplementary Fig. 4). Differences in bacterial community composition (beta diversity) between mice receiving FMT from antibiotic-treated and control infants were evaluated across the seven sampling time points. Mice that received FMT from antibiotic-treated infants presented distinct distribution patterns of the samples over time (as seen by separate clustering), from day 3 to day 43, compared to mice that received FMT from age-matched control infants (Fig. 4d, e; Supplementary Fig. 4c, d). While slight differences after FMT from antibiotic-treated and control infants were already seen at day 3, they became more pronounced with time up to 43 days after FMT (Supplementary

Fig. 5). Moreover, more than 80 altered features in gut microbiome composition were detected 43 days after FMT between mice receiving feces collected at 1 month of age from infants exposed and not exposed to neonatal antibiotics (Supplementary Fig. 7).

**Discussion**

In this work, we report the long-term effects of antibiotic exposure in the neonatal period and later in infancy and childhood on child growth. In a large unselected birth cohort, a significant reduction in weight, height, and BMI Z-scores throughout the first six years of life was observed in boys but not in girls treated with antibiotics during the first days of life. This observation was replicated in an independent cohort, in which neonatal antibiotic exposure was associated with reduced weight and height Z-scores over the first five years of life in boys but not in girls. It is of note that the analyses of both cohorts were adjusted for potential confounding factors including gestational age, maternal pre-pregnancy BMI, mode of delivery, intrapartum antibiotic use, and birth weight Z-scores. Furthermore, the study design of the SFBC cohort allowed us to differentiate between the contributions of neonatal antibiotic exposure and concomitant infection, which is a considerable confounding factor in epidemiological studies attempting to elucidate the long-term effects of antibiotic exposure. Significantly reduced growth was evident in boys who received empirical antibiotics for suspected infection but in whom infection was ruled out. The growth impairment appeared to be somewhat more pronounced in neonates receiving a full course of antibiotics as compared to those who received a shorter empirical treatment. This may be interpreted to imply that neonatal infection might also disturb later growth or, alternatively, suggest a causal dose-response relationship since the duration of antibiotic exposure was significantly longer in neonates diagnosed with infection (Table 1).

The potential causal link between neonatal antibiotic exposure and impaired childhood growth may be mediated by antibiotic-induced perturbations of the developing intestinal microbiome. Intestinal microbes reportedly play essential roles in the digestion of dietary compounds and modulate intestinal energy harvest[19] as well as host energy metabolism and satiety[20]. In the present study, a significantly altered gut microbiome characterized by disorganized development, increasing richness, and particularly reduced abundance and diversity of *Bifidobacterium* species was detected in subjects exposed to antibiotics in the first days of life. These results from a limited number of infants are consistent with previous reports[21–23], which have indicated a reduced abundance of Bifidobacteria up to the age of 90 days in infants exposed to neonatal antibiotics. The present study extends these data with fecal sample follow-up until 24 months of age at which point a significant reduction in Bifidobacteria was still evident. These data may be of clinical importance in light of recently published data suggesting that the specific individual gut microbiome and corresponding metabolomic profiles consolidate during the first two years of life[24]. Our data, therefore, suggest that neonatal

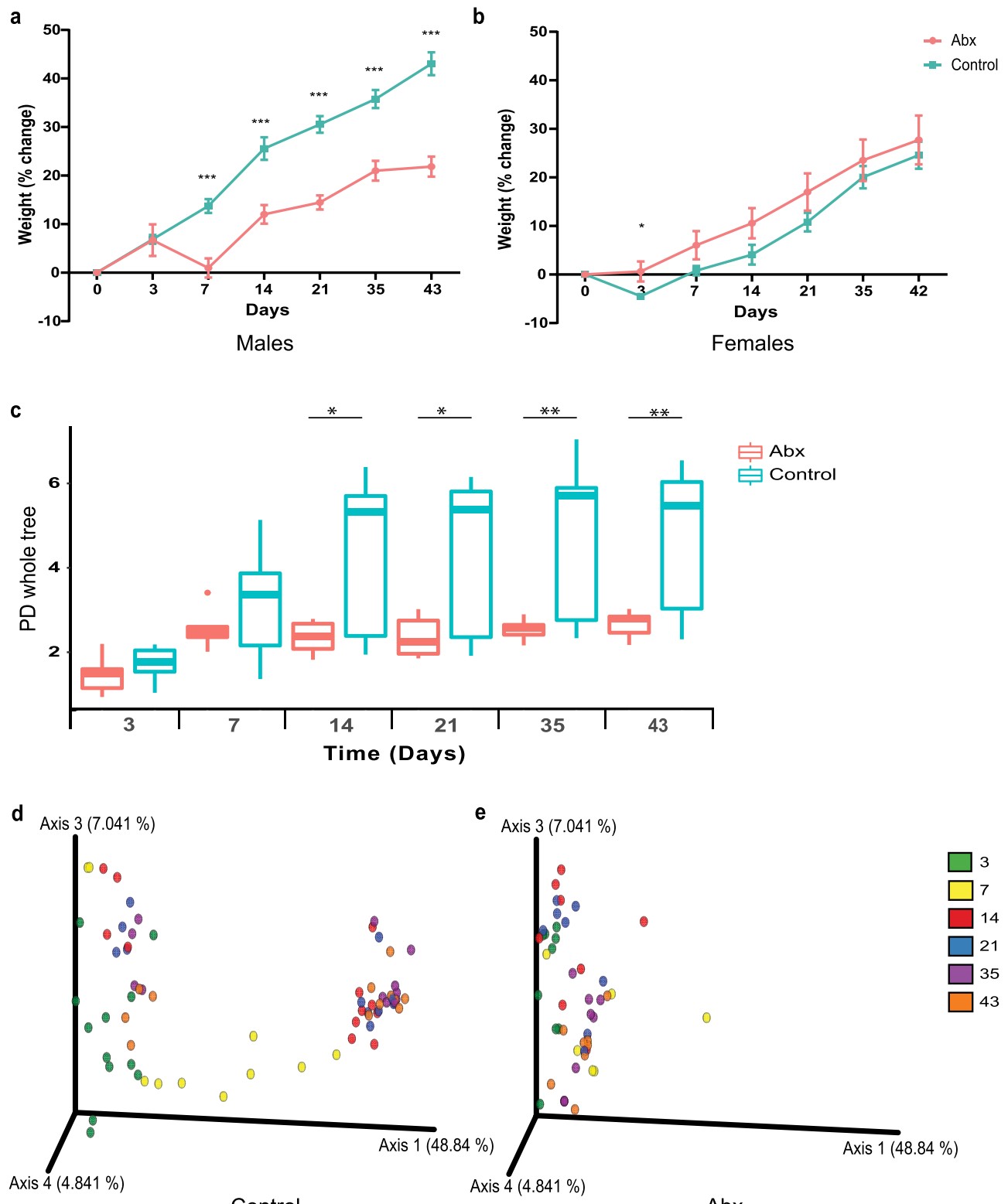

antibiotic exposure has a more profound and long-lasting impact on gut colonization than previously thought. In support of this notion, FMT with fecal samples from antibiotic-exposed infants resulted in intestinal microbiome development abnormalities, which became increasingly pronounced over the 43-day study period in germ-free mice.

The impact of early-life antibiotic use on growth may be dependent on the age at which the exposure occurs. In the present study, neonatal antibiotic exposure was associated with reduced weight and height gain in boys whilst antibiotic use later in infancy and childhood was associated with increased weight in boys and increased BMI in both boys and girls during the first six years of life. We interpret our results to suggest that antibiotic exposure during the neonatal period may be causally related to reduced growth but antibiotic use later in infancy and childhood may have the opposite effect and increase the risk of obesity or

**Fig. 4 Fecal microbiota transplant (FMT) from antibiotic-exposed neonates induces changes in weight gain and bacterial composition in germ-free mice.** (**a**) Male mice receiving FMT from 1-month old infants exposed to neonatal antibiotics (pink) gained significantly less weight compared to mice receiving FMT from non-exposed infants (blue) starting from day 7 (***$p \leq 0.001$). (**b**) Female mice receiving FMT from antibiotic-exposed neonates (pink) exhibit only a transient difference in growth 3 days after transplantation ($p = 0.02$). (**c**) Fecal microbiota alpha diversity comparison based on phylogenetic diversity. Significant differences in bacterial richness between mice receiving FMT from control and antibiotic-exposed infants (day 14 $p = 0.04$; day 21 $p = 0.013$; day 35 $p = 0.004$; day 43 $p = 0.01$). PCoA based on Unweighted UniFrac distances in mice receiving FMT from control (**d**) antibiotic-exposed (**e**) infants at six-time points, 3 (green), 7 (yellow), 14 (red), 21 (blue), 35 (purple) and 43 (orange) days after FMT. (For (A) control group, n = 8–9; abx group, $n = 12$. For (**c–e**), control group, day 3,14,21,35 and 43, $n = 12$; day 7, $n = 11$. Abx group, day 3, $n = 9$; day 7, $n = 6$; day 14,21,35 and 43, $n = 8$. Four samples from control infants and 3 samples from antibiotic-treated infants were used for fecal transplantation). (*$p < 0.05$, **$p < 0.01$, ***$p < 0.001$ and ****$p < 0.0001$; weight data represent the mean ± SEM of at least four samples in each group).

becoming overweight. Consistent with our results, antibiotic exposure in the first days of life has been observed to be associated with reduced growth during the first year of life in a study of 436 term infants[13] whereas antibiotic use after the neonatal period has been linked to excessive childhood weight gain particularly in boys[8,9,14], albeit not all studies have corroborated these results[25].

The contribution of aberrant early gut colonization to disturbances in growth is supported by observations from clinical studies, according to which the composition of the gut microbiome and particularly reduced numbers of Bifidobacteria in infancy are associated with later development of overweight and obesity[26,27]. On the other hand, the delayed gut microbiome maturation encountered in undernourished children is reportedly causally related to the concomitant growth restriction[28]. These data may be interpreted to suggest that the timing and type of antibiotic exposure might be decisive in determining the impact of antibiotic exposure to gut microbiome development and child growth. Neonatal antibiotic exposure was not associated with later antibiotic purchases in the present study (Table 1) and the association between antibiotic use after the neonatal period and increased childhood BMI Z-scores was independent of neonatal antibiotic exposure. Consequently, we interpret our results to reflect an independent link between neonatal antibiotic exposure and later growth. This notion is consistent with data from experimental studies, which showed low-dose penicillin exposure leads to weight gain[10], whereas broad-spectrum[11] or high-dose[29] antibiotic treatment results in impaired growth in mice.

Evidence suggesting a causal relationship between neonatal antibiotic exposure, the ensuing long-term perturbations in gut colonization, and the observed growth impairment in boys was provided by the data demonstrating reduced growth in male but not in female mice receiving FMT from antibiotic-exposed subjects as compared to those receiving FMT from non-exposed children. It is of note that impaired mouse growth was observed after FMT with fecal samples obtained even 24 months after antibiotic exposure. Previously, Blanton and colleagues reported that FMT from children exhibiting gut microbiome immaturity and growth failure caused by undernourishment leads to significantly reduced growth in mice[28]. We interpret our data to suggest that the observed defect in gut microbiome maturation after neonatal antibiotic exposure may carry long-term physiological consequences by impairing height and weight gain during the first six years of life.

Our study has considerable limitations, which warrant restraint in interpretation of the results. The impact of antibiotic exposure in neonates without any symptoms, signs, or risk factors for infection remains unknown for obvious practical and ethical reasons. While the detailed data from the SFBC enabled us to differentiate between neonates briefly exposed to experimental antibiotic treatment and those who were diagnosed with infection by the clinician, it is possible that our results have been confounded by the underlying causes leading to antibiotic exposure also in the subjects in whom infection was ruled out.

It is well-established that breast milk is the most important determinant of infant gut microbiome composition and function and particularly affects specific Bifidobacteria[30–32]. Data regarding breastfeeding was not available for the SFBC cohort and it is possible that differences in the rates and duration of breastfeeding may explain the observed gut microbiome alterations and reduced abundance of Bifidobacteria in subjects exposed to neonatal antibiotics. However, shorter breastfeeding has been associated with higher weight in infancy[33] and childhood[34], and we, therefore, believe that confounding by breastfeeding duration is not very likely. Furthermore, no differences in breastfeeding rates were detected between infants with infection who were exposed to antibiotics in the neonatal period and infants without neonatal infection at any point during the first six months of life in the PEACHES cohort. It is also of note that the decrease in weight and height during the first five years of life in boys with neonatal infection and antibiotic treatment in the PEACHES cohort remained statistically significant after adjusting for breastfeeding. All infants included in the gut microbiome analyses and FMT experiments were breastfed at 1 month of age, which also reduces the possibility of confounding at that time point.

Our results suggest that neonatal antibiotic administration may have detrimental consequences. Impaired childhood growth is known to be associated with poor neurodevelopmental outcomes[35,36] and increased occurrence of cardiometabolic risk factors in later life[37,38]. Our results corroborate previous studies[8,9,14] indicating an association between antibiotic use in infancy and early childhood and increased BMI and thus suggests that the effect of antibiotics on child growth is dependent on the age at which the exposure occurs. The impact of neonatal antibiotic exposure on long-term health warrants detailed study with both epidemiological and experimental approaches. In the clinical setting, rapid means of accurately identifying neonates with bacterial infection are needed to limit antibiotic use in these vulnerable subjects.

## Methods

**The Southwest Finland Birth Cohort.** The epidemiological study is based on the Southwest Finland Birth Cohort (SFBC), which consists of all 14,946 children born in the hospital district of Southwest Finland during the years 2008–2010. Subjects born at full term (after 36[6/7] weeks of pregnancy) from singleton pregnancies were included in the present study. If a woman had more than one pregnancy and delivery during the study period, only the first child was included in this study to ensure independence of the study subjects. Furthermore, chronic disease affecting growth such as genetic syndromes, significant congenital heart disease, malignancies or endocrine, and growth disturbances requiring growth hormone therapy served as criteria for exclusion from the study. Altogether 12,422 children were included in the present study.

Data regarding neonatal antibiotic exposure and diagnoses of neonatal bacterial infections during the first 14 days of life were extracted from the hospital records. As per the protocol of the neonatal unit at the Turku University Hospital throughout the duration of the study, antibiotic therapy was rapidly initiated in all neonates with symptoms or signs suggesting early-onset sepsis and the initial empirical antibiotic therapy in neonates with suspicion of early-onset sepsis

consisted of a combination of intravenous benzylpenicillin and gentamicin. This policy was accepted by the Head of Neonatology at the Turku University Hospital. The children were grouped as follows: (1) empirical antibiotic therapy, which was discontinued after infection had been ruled out, (2) antibiotic therapy for confirmed or clinical infection or (3) no neonatal antibiotic exposure. Of the 638 infants diagnosed with neonatal bacterial infection, 7 (1.1 %) had blood culture positive sepsis, 396 (62%) were deemed to suffer from clinical sepsis, 204 (32%) had neonatal pneumonia and 31 (4.9%) had other bacterial infections. Data regarding pre- and perinatal characteristics of the study subjects and their mothers including gestational age, birth weight, mode of delivery, maternal prepregnancy body mass index, intrapartum antibiotic treatment, time of commencement and duration of antibiotic therapy and antibiotic prescriptions during the first six years of life were extracted from the hospital records and national registers. Birth weight Z-scores were calculated using the recently published references specific to the Finnish population[39]. Growth data were obtained from municipal well-baby clinics. Data on antibiotic purchases during the first six years of life were extracted from the Drug Prescription Register maintained by the Social Insurance Institution of Finland. All data regarding the children in the cohort were collected from national and local registers. No intervention or contact with the families was included in the study and, therefore, no informed consent was required. The cohort/register study was found ethically acceptable and approved by the Finnish Institute for Health and Welfare, a national expert agency under the jurisdiction of the Finnish Ministry of Social Affairs and Health.

The anthropometric data closest to the time points of 6 months and 1, 2, 3, 4, 5, and 6 years of age were used in the analyses. The Finnish growth charts[40] were used to obtain population-specific Z-scores for height, weight, and BMI. The association between neonatal antibiotic exposure and weight and height Z-scores at 6 months and 1 year and weight, height and BMI Z-scores at 2, 3, 4, 5, and 6 years of age were analyzed using hierarchical linear mixed model for repeated measurements using the "MIXED" procedure of SAS, version 9.4 (SAS Institute, Cary, NC, USA). Neonatal antibiotic exposure, duration of pregnancy, birth weight Z-score, mode of delivery (vaginal or caesarean section delivery), time, maternal prepregnancy body mass index, and intrapartum antibiotic treatment (yes or no) were included in the model as explanatory variables. Interaction between neonatal antibiotic exposure and time was included in the model to examine whether mean change over time was different between antibiotic exposure groups. An unstructured covariance pattern was used for repeated measures. Normal distribution assumption was checked from studentized residuals. The data were analyzed separately for girls and boys. The same approach was used to analyze the association between the number of antibiotic purchases after the neonatal period and child growth during the first six years of life. The models contain the same responses. The number of antibiotic purchases, neonatal antibiotic exposure, gestational age, birth weight Z-score, mode of delivery (vaginal or caesarean section delivery), and maternal prepregnancy body mass index were included in the model as explanatory variables.

*The PEACHES cohort.* The PEACHES cohort consists of 1707 offspring and their mothers who were recruited during pregnancy in 23 obstetrics and gynecology departments in Bavaria (southern Germany), the University Hospital of Düsseldorf (western Germany), and parts of northern Germany at 4–6 weeks before their due date[15,16]. The study investigates the long-term effect of pre-conception maternal obesity on the development of overweight and associated metabolic diseases in both mothers and their offspring. The inclusion criteria were maternal age ≥18 years, singleton pregnancy, gestational age at birth of ≥37 weeks, and maternal prepregnancy BMI ≥ 30 kg/m$^2$ or 18.5–24.9 kg/m$^2$ (control group). Women with a preexisting diagnosis of type 1 or type 2 diabetes or other chronic diseases before pregnancy and giving birth to offspring with malformations and/or genetic defects were excluded. For confirmation within the PEACHES cohort, we used R, version 3.6.0, (https://cran.r-project.org/) and the package nlme to perform a linear mixed-effects analysis of the relationship between neonatal infection treated with antibiotics and weight/height/BMI Z-scores together with the covariates child's age, gestational age, mode of delivery (vaginal or caesarean section delivery), maternal prepregnancy BMI, intrapartum antibiotic treatment (yes/no), birth weight Z-score and any breastfeeding (full or partial) during the first 6 months of life. These variables are included as fixed effects in the model, together with the interaction term between neonatal infection treated with antibiotics and child's age. As random effects, we modeled the intercept for subjects as well as by-subject slope for the effect of child's age. We assumed an unstructured covariance pattern for the repeated measures. Weight and height Z-scores were analyzed at 6 months and 1 year of age; weight, height, and BMI Z-scores at ages 2, 3, 4, 5 years. The PEACHES cohort study was approved by the ethics committee of the Ludwig-Maximilians-Universität München, Germany (protocol no. 165–10). Written informed consent was obtained from the participants.

**Infant fecal samples.** The fecal samples for microbiota analyses were collected during a randomized controlled trial assessing the efficacy of maternal probiotic supplementation in reducing the risk of atopic eczema in the infant (NCT00167700) conducted at the Turku University Hospital in Turku, Finland[17]. The clinical trial was a part of a series of prospective studies investigating the health implications of early host-microbe interaction, which included prospective gut microbiota sampling for the assessment of the impact of early exposures on gut

microbiota composition and function. During the course of the clinical study, fecal samples were obtained from all participants at the ages of 1, 3, 6, 12, and 24 months for gut microbiota analyses. Altogether 13 infants exposed to neonatal antibiotics and 20 non-exposed infants were selected for this study based on sample availability. The samples were maintained at −80 °C and shipped on dry ice to the Azrieli Faculty of Medicine, Bar Ilan University, Safed, Israel. Antibiotic-free subjects were selected as controls. The study was approved by the ethics committee of the Intermunicipal Hospital District of Southwest Finland. Oral and written informed consent was obtained from all the families upon enrollment in the clinical study.

**Experimental animals.** Germ-free (GF) Swiss Webster mice were obtained from Taconic Farms Inc., (Germantown, NY, USA) and maintained at the animal facility at the Azrieli Faculty of Medicine. GF mice that underwent fecal transplantat were housed in the conventional animal facility in regular cages and under standard housing protocols. All animals were housed under 12 h light–12 h dark regime and had free access to food and water; all mice were fed from the same food batch (Harlan-Teklad). The experiment was performed using protocols approved by the Bar Ilan University Animal Studies Committee.

**Fecal microbiome transplantation.** Fecal samples from each infant were transplanted to GF mice by oral gavage at 5 weeks of age. Each sample was suspended in 800 µl of sterile phosphate-buffered saline (PBS) and dissolved by vortex for 1-min. A total of 200 µl of the fecal suspension was administered by oral gavage to one to three GF mice. The process took place once, immediately after the mice were taken out of the isolator. To minimize cage effects, each treatment group was housed in at least two cages, while mice transplanted with the same sample were not housed in the same cage. Mice were followed for 43 days, and day 0 refers to the day of the fecal transplantation. Stool microbiome and weight was examined at seven-time points (days 0, 3,7,14,21,35,43). Stool samples were stored at −80 °C.

**Bacterial DNA extraction, amplification, and sequencing.** DNA was extracted from infant and stool samples using the Power Soil DNA Isolation Kit (MoBio) according to the manufacturer's instructions, using a Beadbeater (BioSpec) for 2 min. Following DNA extraction, the V4 variable region of the bacterial 16 S rRNA gene was amplified by polymerase chain reaction (PCR) using the 515 F and 806 R primers, and each sample received a unique 515 F barcoded primer, in order to identify each sample during data analysis. Primer sequences used were: 515F-(barcode) 5′-AATGATACGGCGACCACCGAGATCTACACGCTAGCCTT CGTCGCTATGGTAATTGTG TGYCAGCMGCCGCGGTAA-3′ and 806 R 5′- C AAGCAGAAGACGGCATACGAGATAGTCAGTCAGCCGGACTACHVGGGT WTCTAAT -3′[41]. PCR reactions were carried out with the Primestar taq polymerase (Takara) for 30 cycles of denaturation (95 °C), annealing (55 °C), and extension (72 °C), and a final elongation at 72 °C. Products were purified using AMPure magnetic beads (Beckman Coulter), and quantified using Pico-green dsDNA quantitation kit (Invitrogen). Samples were pooled at equal concentrations (50 ng/µl), loaded on 2% E-Gel (Thermo Fisher), and purified using NucleoSpin Gel and PCR Clean-up (Macherey-Nagel). Purified products were sequenced using the Illumina MiSeq platform (Genomic Center, Azrieli Faculty of Medicine, BIU, Israel). For the shotgun samples, libraries were prepared with the NexteraXT DNA Library Preparation Kit (Illumina) and sequenced on the HiSeq2500 and NovaSeq machines (Illumina). We sequenced the 27 samples for a total of 76.9 Gb (2.8 Gb average per sample after quality control, 1.1 Gb standard deviation). The raw reads were submitted to the NCBI-SRA archive and are available under BioProject PRJNA606271.

**Normalization.** Given the large variation in feature values, we transformed these values to Z-scores by adding a minimal value to each feature level (0.01) and calculating the log10 of each value. Statistical whitening was then performed on the table, by subtracting the average and dividing by the standard deviation of each feature. We then repeated the process for each patient.

**16S rRNA gene sequence analysis.** FASTQ data was processed and analyzed using Quantitative Insights into Microbial Ecology 2 (QIIME2) pipeline version 2018.2[42]. Single-end sequences were first demultiplexed using the barcode sequences. In order to improve taxonomic resolution, reads were denoised and clustered using DADA2[43], primers were trimmed off and single-end read were truncated to ≥150 base pairs. For taxonomy classification, final feature sequences were aligned against Greengenes database[44] with 99% confidence. In order to avoid any possible contamination, the infant feature table was generated by filtering features which were not found in 40% of each sample group, and for the mouse feature table, features with <0.001% frequencies in total were removed.

The analysis, for both infants and mice, was performed on a rarefied table of >10,000 reads per sample after filtration of mitochondria and chloroplast sequences. Alpha diversity was calculated using the Faith's Phylogenetic Diversity[45] (PD Whole tree) measure, referring to bacterial richness within the sample, while significant differences in bacterial richness between the groups were generated using the Kruskal–Wallis test. Beta diversity was analyzed according to weighted

(quantitative) and unweighted (qualitative) UniFrac[46] distances in order to compare differences in gut bacterial communities between the sample groups. To evaluate the level of significance, a permutational multivariate analysis of variance (PERMANOVA) test was performed, as implemented in QIIME2 with the default of 999 permutations for both weighted and unweighted UniFrac.

Regarding the mouse experiment, evaluation of significant differences in bacterial abundance between Abx and control transplanted groups, at all taxonomic levels, were done using the discrete false-discovery rate (DS-FDR) test according to the last day of the experiment (Day 43). dsFDR was calculated using the mean-rank test statistics with an FDR threshold of 0.05[47]. The analysis and Heatmap visualization was created using Calour[48].

**Statistical analysis**. Mouse body weight was normalized according to day 0, the first day of the experiment, in order to calculate changes between Abx and control transplanted mouse groups over time. Differences in weight gain fold change were assessed using unpaired two-tailed t-test. All weight data are mean ± SEM. Unsupervised Learning was performed on the rarefied and subsampled version of the 16 S rRNA feature table in order to recognize patterns in the data. Principal Component and Principal Coordinates Analysis were performed using Python version 3.5, and its sklearn package. A 2-tailed $p$-value of less than 0.05 was taken to indicate statistical significance. Asterisks indicate significance ($*p < 0.05$, $**p < 0.01$, $***p < 0.0001$).

**Metagenomics analysis of community composition and *Bifidobacterium* species**
*Data assembly.* Twenty-seven metagenomics samples collected from 20 infants at the ages of 1 and 6 months were analyzed (Table S5). Samples with low sequencing quality were trimmed using Trimmomatic[49] v0.39 with parameters ILLUMINA-CLIP:NexteraPE-PE.fa:2:30:10 LEADING:3 TRAILING:3 SLIDINGWINDOW: 4:15 MINLEN:36. Each sample was assembled separately using megahit v1.1.3[9], with parameter -presets meta-sensitive with only scaffolds longer than 500 bp used for later analysis. To evaluate the fraction of reads that went into the assemblies and to calculate coverage for scaffolds we used bowtie2 v2.3.2[10] with parameters -sensitive –X 1000 -reorder. Total assembly size for the different samples ranges from 11–75 Mbp in scaffolds >500 bp long, with 93–99% of the reads mapped to the assembly. The high mapping rate suggests that our assemblies cover most of the genetic material in the samples.

*Genome recovery for* Bifidobacterium *species.* To reconstruct bins for *Bifidobacterium* species we applied taxonomy- and differential-coverage based binning approaches[50]. For each sample, we used a semi-automatic procedure for genome recovery as follows. First, all genes in scaffolds longer than 4000 bps from each sample were aligned (using blast[51], parameters -dust no - soft_masking false) against a database of *Bifidobacterium* genes collected from 456 Bifidobacterium strains taken from NCBI's RefSeq database. Next, all scaffolds with a significant number of *Bifidobacterium* hits were assigned taxonomy based on the species with the highest number of hits. These scaffolds were further manually binned by plotting their assigned taxonomy, coverage, and %G + C using R. This approach allowed us to identify scaffolds that belong to the same genome but received different taxonomic assignments from closely related species. It also enabled us to identify different strains, with different coverages, for the same species. Taxonomy was assigned to the bins based on the most frequent hit to their genes. Completeness and contamination were evaluated using CheckM[13]. Overall, we reconstructed 54 *Bifidobacterium* genomes of which 36 are estimated to be >90% complete and <5% contaminated. Table S5 summarizes the information about the recovered genomes.

*Evaluating community composition from metagenomics samples.* We used metaphlan2 v2.6.0[14] to evaluate community composition. 5.5 million reads were used from each sample; species whose abundance was >0.1% were considered. Figures were created using R.

## Data availability
All 16 S rRNA data are available at EBI: project ID ERP121888 [https://www.ebi.ac.uk/ena/browser/view/PRJEB38457]. The metagenomic raw reads were submitted to the NCBI-SRA archive and are available under BioProject PRJNA606271. Source data are provided with this paper.

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

## Acknowledgements

We would like to acknowledge Eliisa Löyttyniemi, MSc, for statistical consultation and Dr. Ulla Sankilampi, MD, PhD, for providing the algorithms for anthropometric Z-score calculations. The authors are also grateful to all the families who took part in this study. A grant from the Finnish Society for Pediatric Research (SR). Grants for the PEACHES cohort from the German Federal Ministry of Education and Research and the Foundation for Cardiovascular Prevention in Childhood, Ludwig-Maximilians-Universität München, Munich, Germany (RE).

## Author contributions

E.I., S.S., S.R., and O.K. conceptualized the study. A.U., O.T., A.B., O.Z., H.N., E.P., A.O., H.B., H.K., H.O., A.K., N.S., I.S., H.L., Y.L., C.K., S.P., R.E. analyzed the data. A.U., O.T., H.N., C.K., S.P., R.E., S.R., and O.K. wrote the manuscript.

## Competing interests

The authors declare no competing interests.
