## [Peer Review File · Nature Communications]

REVIEWERS' COMMENTS

Reviewer #1 (Remarks to the Author):

The authors responded well to my comments at the prior journal. Including the new data is helpful. My only remaining comment is about the interpretation of the data. There are 2 findings: lower weight trajectory in children with neonatal antibiotic exposure, and high trajectory if the exposure is somewhat later. In both cases, the trajectories are abnormal. To me, that is the main story—that antibiotics affect metabolism but the direction of the changes is based on when in early life the exposure occurs. I recommend changing the title of the paper, and the relevant sections to reflect the author's findings. This is an important observation that has both pathophysiological and clinical implications.

Martin Blaser

Response to the remarks by Reviewer #1

The authors responded well to my comments at the prior journal. Including the new data is helpful. My only remaining comment is about the interpretation of the data. There are 2 findings: lower weight trajectory in children with neonatal antibiotic exposure, and high trajectory if the exposure is somewhat later. In both cases, the trajectories are abnormal. To me, that is the main story—that antibiotics affect metabolism but the direction of the changes is based on when in early life the exposure occurs. I recommend changing the title of the paper, and the relevant sections to reflect the author's findings. This is an important observation that has both pathophysiological and clinical implications.

Martin Blaser

Response: We thank Dr. Blaser for his kind and encouraging remarks. We have changed the title as follows:

Neonatal antibiotic exposure impairs child growth during the first six years of life by perturbing intestinal microbial colonization while antibiotic exposure later in childhood is associated with increased body-mass index

We leave the decision whether to use the new longer title or the original to the Editor's discretion.

In addition, the following additions and changes have been made in the manuscript to reflect the importance of the timing of antibiotic exposure in the effect on child growth:

Abstract:

Exposure to antibiotics in the first days of life is thought to affect various physiological aspects of neonatal development. Here, we investigated the long-term impact of antibiotic treatment in the neonatal period and early childhood on child growth in an unselected birth cohort of 12,422 children born at full term. We find significant attenuation of weight and height gain during the first 6 years of life after neonatal antibiotic exposure in boys, but not in girls, after adjusting for potential confounders. In contrast, antibiotic use after the neonatal period but during the first six years of life is associated with significantly higher body mass index throughout the study period in both boys and girls. Neonatal antibiotic exposure is associated with significant differences in the gut microbiome, particularly in decreased abundance and diversity of fecal Bifidobacteria until 2 years of age. Finally, we demonstrate that fecal microbiota transplant from antibiotic-exposed children to

germ-free male, but not female mice results in significant growth impairment. Thus, we conclude that neonatal antibiotic exposure is associated with a long-term gut microbiome perturbation and may result in reduced growth in boys during the first six years of life while antibiotic use later in childhood is associated with increased body mass index.

Page 3, line 18:

Later in infancy and childhood, antibiotic use has been linked to increased risk of overweight and obesity^{8,9,12}. We hypothesized that antibiotic treatment during the first days of life may exert a long-lasting effect on childhood growth by disrupting the natural gut microbial colonization process.

In this work, we investigated the association between antibiotic exposure during and after the neonatal period and child growth until the age of six years in a large unselected cohort. We show that neonatal antibiotic exposure is associated with reduced weight and height gain during the first six years of life in boys but not in girls. After the neonatal period, however, antibiotic use is associated with increased body mass index (BMI) in both boys and girls. Experimental studies using germ-free mice demonstrate that fecal microbiota transplant (FMT) with fecal samples from infants exposed to antibiotics in the neonatal period results in growth failure in male but not in female mice, which suggests a causal role for the antibiotic-induced gut microbiome perturbations in the pathogenesis of the growth impairment in antibiotic-exposed boys.

Page 9, line 23:

In this work, we report the long-term effects of antibiotic exposure in the neonatal period and later in infancy and childhood on child growth.

Page 11, line 1:

We interpret our results to suggest that antibiotic exposure during the neonatal period may be causally related to reduced growth but antibiotic use later in infancy and childhood may have the opposite effect and increase the risk of obesity or overweight. Consistently with our results, antibiotic exposure in the first days of life has been observed to be associated with reduced growth during the first year of life in a study of 436 term infants¹³ whereas antibiotic use after the neonatal

period has been linked to excessive childhood weight gain particularly in boys^{8,9,14}, albeit not all studies have corroborated these results²⁷.

Page 12, line 29:

Our results corroborate previous studies^{8,9,14} indicating an association between antibiotic use in infancy and early childhood and increased BMI and thus suggests that the effect of antibiotics on child growth is dependent on the age at which the exposure occurs.

Samuli Rautava

Omry Koren